# “I Had to Rediscover Our Healthy Food”: An Indigenous Perspective on Coping with Type 2 Diabetes Mellitus

**DOI:** 10.3390/ijerph19010159

**Published:** 2021-12-24

**Authors:** Maya Maor, Moflah Ataika, Pesach Shvartzman, Maya Lavie Ajayi

**Affiliations:** 1Department of Sociology and Anthropology, Ariel University, Ariel 4070000, Israel; 2Clalit Health Services, Siaal Research Center for Family and Primary Care, Division of Community Health, Ben Gurion University of the Negev, Beer-Sheva 8410501, Israel; ataikam@bgu.ac.il; 3Pain and Palliative Care Unit, Siaal Research Center for Family Medicine and Primary Care, Division of Community Health, Ben Gurion University of the Negev, Beer-Sheva 8410501, Israel; pshvartzman@gmail.com; 4Gender Studies, Ben Gurion University of the Negev, Beer-Sheva 8410501, Israel; laviema@bgu.ac.il

**Keywords:** diabetes, the Bedouin community, social inequality, active coping

## Abstract

Type 2 Diabetes Mellitus (T2DM) is disproportionally prevalent among the Bedouin minority in Israel, with especially poor treatment outcomes compared to other indigenous groups. This study uses the perspective of the Bedouins themselves to explore the distinct challenges they face, as well as their coping strategies. The study is based on an interpretive interactionist analysis of 49 semi-structured interviews with Bedouin men and women. The findings of the analysis include three themes. First, physical inequality: the Bedouin community’s way of coping is mediated by the transition to a semi-urban lifestyle under stressful conditions that include the experience of land dispossession and the rupture of caring relationships. Second, social inequality: they experience an inaccessibility to healthcare due to economic problems and a lack of suitable informational resources. Third, unique resources for coping with T2DM: interviewees use elements of local culture, such as religious practices or small enclaves of traditional lifestyles, to actively cope with T2DM. This study suggests that there is a need to expand the concept of active coping to include indigenous culture-based ways of coping (successfully) with chronic illness.

## 1. Introduction

Type 2 diabetes mellitus (T2DM) is disproportionally prevalent among indigenous groups, where its treatment outcomes are especially poor, and the complication rates are higher. This paper is based on the first qualitative study that is designed to explore the narratives of coping with T2DM by the Bedouin men and women from Israel. On the basis of a critical analysis of the literature in the field, we present a typology of the main theoretical approaches to the study of coping with T2DM among indigenous minorities: the biomedical approach, the minorities in transition approach, and the social justice approach. We explain the fundamentals of each approach and the reason behind our choice to employ the social justice approach in the analysis of our findings.

Following the presentation of the study’s methodology, we demonstrate how listening to people’s own narratives expands the previous understandings of indigenous culture in at least two ways. First, the influence of indigenous culture on coping is mediated through ethnic inequality, such as a lack of routine medical examinations or unsuitable living conditions. Thus, the negative influence of indigenous culture on coping is not predetermined. Second, indigenous narratives allow us to see how indigenous culture may be used in the service of optimal coping, e.g., using religious beliefs to counter T2DM-related stigmas or using traditional indigenous foods to control glucose blood levels. Together, these findings elucidate the critically important role of qualitative in-depth interviews in both explaining and targeting obstacles specific to indigenous groups that cope with T2DM, as well as utilizing indigenous groups’ knowledge as resources for coping.

### 1.1. T2DM among Bedouins in Israel

Type 2 diabetes mellitus (T2DM) is the result of insulin resistance and is caused by the failure of the body’s cells to use insulin properly, at times combined with an absolute insulin deficiency. It is the most common disease that has become a significant public health concern worldwide [1]. T2DM is associated with a series of significant health complications, as well as a significant economic burden [1,2]. Since some of the damage to health starts at a prediabetic stage, and since treating T2DM once it is diagnosed does not usually restore normal blood glucose levels, screening individuals at risk, and early diagnoses, are of crucial importance [1,2]. T2DM is closely associated with lifestyle choices, such as diet and exercise habits [3,4]. Social and cultural factors are dominant in the etiology, prevalence, and prognosis of T2DM, since the so-called lifestyle choices are often mediated by cultural and social contexts [2]. After a diagnosis, the modification of the individual’s lifestyle, preferably through accessible and culturally relevant lifestyle modification programs, has been consistently found to be the best indicator of positive treatment outcomes [1,2]. Thus, understanding the complex interactions of culture and diabetes is of pivotal importance in the prevention, early detection, and treatment of T2DM.

Moreover, the interaction of culture and T2DM is even more significant, as T2DM is disproportionally prevalent among ethnic minorities and indigenous groups, e.g., African Americans [5] and Native Americans [6,7]. The high prevalence of diabetes among these groups is accompanied by especially poor treatment outcomes and higher complication rates in comparison to other social groups [5,7]. Poor treatment outcomes commonly lead to health complications such as kidney disease, blindness, limb amputation, nerve damage, heart disease, and cerebrovascular incidents [8].

The current study focuses on the Bedouin community of the Negev region of southern Israel. The Bedouin are a subset of the minority Arab community of Israel, and until relatively recently, they were a nomadic people. Since the establishment of the State of Israel in 1948, the Bedouin community has undergone a process of social and economic marginalization, influenced by a rapid—and at times, forced—transition to a semi-urban way of life [9]. Some moved to government-established townships, others to unrecognized villages that lack basic infrastructure and services, such as electricity, a water supply, educational institutions, and health care services, because the state disputes the legality of these settlements and, consequently, refuses to invest in basic infrastructure. Economically, the unrecognized villages are more disadvantaged than the legally recognized townships and are considered among the poorest settlements in Israel [8]. The transition to a semi-urban way of life has impacted changes in kinship structure, gender relations, economic structure, and political power [9].

T2DM is disproportionally prevalent in the Bedouin population of Israel, mirroring the experiences of indigenous and minority groups elsewhere in the Western world [10,11,12]. For example, a study from 2002 found that prevalence of T2DM among Bedouins in the Negev above the age of 35 years was as high as 7.3% for men and 9.9% for women [13]. A study from 2007 found the age-adjusted diabetes rate among Bedouins in Israel to be 12%, compared to only 8% among Jewish Israelis. The prevalence rate was especially notable among Bedouins between 40 and 49 years of age, where the prevalence rate was three times higher than in the Jewish population of the same age [14].

Treatment outcomes, as defined by the adherence to treatment, are poor, with 73% of Bedouin diabetes patients defined as not adhering to treatment [11]. Results remain poor when measured by diabetes controls, as one study found that only 29.3% of Bedouin patients had diabetes under control, compared to 46.7% among non-Bedouin patients [15].

### 1.2. Different Theoretical Approaches to T2DM among Bedouins in Israel

#### 1.2.1. The Biomedical Approach

Most studies of coping with T2DM within the Bedouin communities in Israel employ the biomedical approach. According to this approach, which frames medical knowledge as objective and value-free, e.g., [8,14,16,17,18], coping and treatment outcomes are under the responsibility of the individual. According to this approach, negative treatment outcomes or complications are the result of the individual’s lack of compliance with medical advice. For example, Tamir et al. assessed adherence to follow-up tests and drug treatments for a range of illnesses, including diabetes, in Bedouin communities in Israel and found that only 28% of diabetes patients complied with drug therapies [14]. They noted that the adherence to drug therapies and follow-up tests among Bedouin patients were alarmingly low in comparison to that of the Jewish Israeli population. At the same time, the authors’ exclusive reliance on the analysis of data taken from medical records meant that they were not able to explore the underlying mechanisms that informed this low rate of compliance.

Studies that adopt the biomedical approach focus on individual factors as a potential explanation for poor coping, which is measured by the lack of adherence to treatment regimens and follow-up tests. These studies do not consider the wider social context as a possible explanation for distinctive coping strategies among the Bedouin. In fact, implementing this medicalized individual approach to understanding coping with diabetes among specific ethnic groups reflects a contradictory approach to ethnic identity. On the one hand, it recognizes that specific groups (e.g., Bedouins in Israel) have different experiences to those of Israel’s ethnic majority in terms of diabetes rates and coping with diabetes. However, on the other hand, this approach does not recognize other forms of differences that may characterize these groups, including structural differences, such as different living conditions, that may make following biomedical advice very difficult. Instead, these approaches assign the responsibility for the possible difficulties in following medical advice to the individuals’ lack of cooperation or their fatalistic approach.

#### 1.2.2. The Minorities in Transition to Modernization Approach

We use the term “minorities in transition to modernization approach” to refer to the approach utilized in a cluster of studies that add an epidemiological focus to the individualized medical approach by specifically addressing the disproportionately high rates of diabetes and poor treatment outcomes among indigenous groups. An analysis of the theoretical underpinnings of this cluster of studies reveals a common narrative. According to the minorities in transition approach, the high prevalence of T2DM and the negative treatment outcomes among ethnic minorities in Western society are the result of a rapid transition from a nomadic agricultural lifestyle to a modern urban lifestyle, e.g., [13,19,20,21].

Like the biomedical approach, this approach is based on the understanding of a biomechanical model of disease, according to which, coping is an individual matter and optimal coping is manifested through an adherence to medical advice. Unlike the biomedical approach, studies that employ the “minorities in transition” approach address not only individual characteristics but also group or social/cultural characteristics, such as practices and beliefs shared by members of a specific community, as factors that may influence coping with T2DM.

Research by Yoel et al. included these two approaches [19]. Reflecting the biomedical rationale, they used questionnaires that accessed personal characteristics in an attempt to understand why some Bedouin patients fail to achieve optimal treatment outcomes. At the same time, the researchers moved beyond the biomedical model when they measured the extent to which Bedouin individuals held specific cultural beliefs that were attributed to the Bedouin community as a ‘traditional community,’ such as a suspicion towards Western medications or an inaccurate understanding of chronic illnesses. The researchers linked the extent to which patients held traditional beliefs and a noncompliance with treatment, and argued that traditional beliefs are the main obstacle to optimal treatment outcomes.

Another example of a study that used the minority in transition approach is the study by Dunton et al. who presented a case study of an influential patriarchal figure in a Bedouin community to demonstrate how traditional social and gender practices, within the wider setting of a rapid transition to modernization, can complicate coping with T2DM [21]. They focused on the social responsibility of the sheik (an honorific title in Arab culture commonly used to designate the ruler of a tribe/community) to host family, friends, and other influential figures in the community. The responsibility of hosting means frequent social gatherings in which the sheik and the guests feel a social obligation to eat foods rich in carbohydrates and sugars, as well as sugary drinks. They concluded their analysis with the claim that “These cultural norms present a challenge to diabetes care and management” [21] (p. 1). However, the researchers ignored the fact that before the disruption of the traditional nomadic lifestyle, sugary foods and drinks were not as central to social gatherings as they are today, and events were part of a lifestyle that included regular physical activity.

Other studies also explored the characteristics of Bedouin culture as a potential complicating factor. For example, in a paper that discussed the cultural modifications needed for an intervention program designed to facilitate coping with T2DM, traditional Bedouin nutrition is described as rich in carbohydrates and an obstacle to maintaining a suitable diet [22].

The minorities in transition approach expands the biomedical approach by exploring not only individual factors, but social practices and norms as well, taking a significant step in shedding light on coping among culturally different (i.e., not hegemonic Western) groups. Nonetheless, there are at least three major criticisms regarding the minorities in transition approach. First, this approach assumes that while Western communities have long been accustomed to a modern lifestyle and have therefore adopted practices and beliefs that balance the damaging effects of modern living conditions, such as the acceptance of biomedical knowledge, biomedical advice, the cooperation with the biomedical system, and self-monitoring, indigenous groups still live according to traditional norms and practices that were developed during the time when they lived nomadic lifestyles. In the future, when the transition to modernity is complete, indigenous groups will also incorporate so-called modern practices and beliefs; therefore, the prevalence of T2DM among them will decrease, and their treatment outcomes will improve.

Hence, this approach considers modern living conditions as the norm. Many studies acknowledge that a modern or urban lifestyle includes many risk factors for T2DM, such as a lack of physical activity or processed foods, but these elements are seen as necessary and inevitable, e.g., [19,21,22,23]. Since modern ways of life are seen as positive and the default in the Western world, traditional (i.e., non-Western) norms and practices are singled out as the problem.

Second, this approach views indigenous culture as having only negative influences on coping. Thus, it does not explore the possibility that elements of indigenous culture can yield effective coping strategies that may not have been taken into account by the biomedical model. For example, some studies acknowledge the need to change the healthcare system and suggest adapting biomedical intervention programs to the specific needs of the Bedouin community, e.g., [22,23]. However, none of these studies have examined coping options that go beyond those offered by the biomedical model, nor have they examined Bedouin culture as a potential source of such options.

Third, this approach does not pay sufficient attention to the economic, political, and social conditions that shape the context in which cultural norms are established. For example, in the case of hospitality and the serving of high carbohydrate foods, hospitality that involves the serving of foods and beverages that are detrimental to the health of diabetics may become more important as traditional social institutions associated with the nomadic lifestyle fade away [21]. In addition, poverty and the intermittent supply of electricity may increase a reliance on high-sugar and high-carbohydrate processed foods in the context of hospitality.

#### 1.2.3. The Social Justice Approach

The fact that indigenous groups tend to be socially, economically, and culturally marginalized, and have a high incidence of poor treatment outcomes, has prompted researchers to explore how specific issues related to the social inequality of these groups may influence their ability to cope. Furthermore, some researchers argue that specific cultural norms and practices can, at least potentially, serve as a positive resource for the coping strategies that expand on those acknowledged by the biomedical approach. These works draw from a previous scholarly transition that shifted from the concept of the adherence to medical advice to the concept of active coping. Active coping is defined as the active and creative balance of medical advice and the fulfilment of personally, culturally, and socially valued activities, goals, and aspirations, while maintaining optimal health [24,25,26]. Since life goals and aspirations, as well as living conditions, are culturally mediated, the transition to active coping also allows researchers to make the connections between individual coping and the broader social, cultural, and ethnic factors.

We use the term “social justice approach” to refer to the approach used in a cluster of studies that explored how active coping is mediated and shaped by various social, economic, and cultural factors. These factors include the structural conditions of social inequality and oppression, which are especially prevalent in the lives of ethnic minorities and indigenous groups. The social justice approach criticizes the biomedical narratives of illness and coping by showing that the biomedical ideas regarding proper coping are often inapplicable to indigenous groups due to their everyday living conditions, and that the biomedical conceptualizations of coping are too narrow, as they rely solely on Western tradition, thus excluding coping strategies that draw on indigenous experience.

We argue that the social justice approach can be very useful in research regarding coping with T2DM in the Bedouin community in Israel, in light of existing structural, social, and health inequalities. The Bedouins are one of the most deprived population groups in Israel [27,28,29]. High poverty rates and poor nutrition can make it more difficult to balance blood glucose levels, purchase prescribed medication, or engage in regular physical activity [20,30]. Indeed, the typical Bedouin meal is low in protein, iron, and calcium content, but is rich in salt and calories [20,30].

In addition, beginning in the 1960s, the State of Israel initiated a process of urbanization and pressured the Bedouins to move into semi-planned towns. Many scholars claim this was done to move the Bedouin out of the Negev in order to allow Jewish Israelis to settle there [31,32]. The forced transition to the semi-planned towns exacerbated the existing social and economic deprivation and produced a social and cultural vacuum [9]. In addition, as of today, parts of the Bedouin community continue to live in unrecognized villages that lack standard government services and sufficient public infrastructure (e.g., water, electricity, public transportation, health services, and education services). Even recognized Bedouins towns and villages are offered significantly fewer healthcare services compared to other Arab and Jewish Israeli areas [33].

In this paper we argue, using the social justice approach, that what is often interpreted as personal or cultural noncompliance with treatment may, in fact, be related to a poor accessibility to medical services, high poverty rates, and other structural inequality factors.

Their similarities with indigenous groups in other parts of the world notwithstanding, the Bedouins in Israel are also characterized by distinct conditions that are not necessarily shared with other indigenous groups. These include increasingly high rates of polygamy [9], the unique structure of the Israeli health system, the specific historical modes of changes in living conditions [34], and a distinct cultural heritage. Thus, given the unique social, geographical, and cultural context of Bedouin groups in the Negev, insights gathered from an empirical study of this population, guided by a social justice framework, can contribute to the theoretical and empirical rigor of the social justice approach.

The goal of the present study is to utilize the social justice approach to understand coping with T2DM among Bedouins in Israel. Specifically, we aim to (1) explore the distinct challenges and barriers the Bedouin face in coping with T2DM and (2) to examine how successful coping can be conceptualized based on the perspectives of Bedouins in Israel. Thus, the section of the results will be divided into two main parts: the barriers to coping with T2DM in the Bedouin community, and the unique resources for coping with T2DM according to the two goals of the study.

## 2. Materials and Methods

The current study used the interpretative interactionism approach [35]. This approach, with its focus on the reciprocal relations between people and their social environments [35], was especially suitable for this study, because “it puts the patient at the center of the research process and makes visible the experiences of patients as they interact with the healthcare and social systems that surround them” [36] (p. 39). This approach includes a number of stages that are carried out as part of a circular process. The first of these is deconstruction, which consists of a critical review of the research literature to identify the preconceptions in the literature and existing tensions in interactions between patients, healthcare providers, and policies. The second is the epiphany, which consists of collecting stories that highlight the meanings attributed by participants to formative life experiences. The third is bracketing, which is analogous to data analyses. The fourth is contextualization, which involves taking into account the social, political, economic, and physical environments of participants, and contextualizing their stories within the wider context of inequality [35,36].

### 2.1. Participants and Recruitment

The interviewees were recruited through a clinic that is part of Clalit Health Services, the largest HMO in Israel, in the Bedouin city of Rahat, as well as through community projects run by the Arab-Jewish Center for Empowerment, Equality, and Cooperation (Negev Institute for Strategies of Peace and Economic Development). The inclusion criteria for participant selection were the self-identification as a Bedouin and having been diagnosed with T2DM for at least 6 months (to ensure a sufficient experience of coping with the illness).

The final study population consisted of 49 interviewees and was varied in terms of gender, age, geographical location, education, and socioeconomic status. After 49 interviews, the researchers determined through reflection on the data collection and analysis process that the saturation point (i.e., no new information was being received from participants) had been reached. Consequently, data collection ceased at this point.

The sample consisted of 34 women and 15 men. The mean age was 57 years, with a range between 18 and 78 years. Many of the interviewees had not completed high school, but several had an academic education. Thirty-six of the interviewees were married (mainly in polygamy marriages), seven were widowers, four were divorcees, and two were single.

### 2.2. Data Collection and Analysis

Ethical approval was obtained from our university’s Human Subject Research Committee, and from the Helsinki Ethics Committee of the Clalit HMO, to allow for the recruitment of participants through different channels. We prepared information sheets in Arabic for distribution to the participants. All participants signed an informed consent form that was written in Arabic.

Qualitative semi-structured interviews were conducted in Arabic in the homes of the participants, or in the clinic, in accordance with the participants’ preference. One female and one male research assistant who spoke both Arabic and Hebrew conducted the interviews. The interviews were based on the narrative approach [37]. Each interview began with a request to hear the illness narrative of the patient (“can you tell me your story of diabetes?”). When the interviewee completed the narration of their story, the interviewer asked Appendix A about signs and symptoms, tests, receiving the diagnosis, telling other people about it, any previous knowledge of diabetes, communication with health professionals, financial impacts, information and support needs, treatment decisions, and the use of complementary approaches.

All audiotaped interviews were transcribed and then translated into Hebrew. A thematic analysis was conducted [38]. The analysis was inductive, in that it relied on the use of emic codes that had emerged from the interviews, and deductive in that it was shaped by a critical literature review [34]. The data analysis was conducted in Hebrew and the quotes used in this manuscript were all translated into English by the authors.

## 3. Results

The analysis of the interviews produced many themes. In this paper, we focus on themes related to two questions: what are the unique barriers to coping with T2DM among Bedouins in Israel that result from physical and social inequalities? What are the unique resources of the Bedouin community for coping with T2DM?

### 3.1. Barriers to Coping with T2DM in the Bedouin Community

#### 3.1.1. Physical Inequality: Land Dispossession

Beginning in the 1960s, the State of Israel initiated a process of urbanization and pressured the Bedouin to move into semi-planned towns. The forced transition to these towns prevented the Bedouin from sustaining a lifestyle based on traditional agriculture. This complicated the ability of patients to cope with T2DM on at least two levels. First, members of the Bedouin community could no longer engage in the daily physical activities involved in traditional agriculture. Second, they lost what is called “food sovereignty” [7]. Without the possibility of engaging in traditional agriculture, traditional foods (such as dried milk products or self-grown fruits and vegetables that are expensive to buy on the open market) were removed from their menu and replaced with processed cheap foods, as the following excerpt demonstrates:

Before, we didn’t have cancer or diabetes and now the Negev is full of diabetics, not because the Bedouin themselves have changed. What has changed is the lifestyle, the state has forced us to change […]. The state took us and placed us in a town [… ]. They changed us. We don’t have lands to grow fruits and vegetables. In the past, we used to grow everything, also poultry and animals. The state fought us in every domain of life. We used to be a society that was based on production and now we’re [a society] based on consumption. This [process] was initiated by the state. Even the fact that women are now inactive and idle is the result of the state changing our lifestyle (Jamila, a 67-year-old woman from a recognized settlement).

Jamila’s words emphasize how state-imposed urbanization has reduced the ability of the community to cope with diabetes. These changes have hindered the ability of people to eat healthy fresh food. Furthermore, Jamila compares women’s current status in the community to their status in the past, where they served a central role in the community thanks to their role in traditional agriculture [9]. In addition to the fact that traditional agriculture is no longer possible, gender norms, a lack of public transportation, and other factors complicate women’s ability to work outside the home, and the result is that they have become physically inactive.

The transition from the production of foods to their consumption has complicated the ability to cope with T2DM in at least two additional ways. First, it has caused a decrease in the daily physical activity involved in traditional agriculture. Second, it has weakened the collective sense of meaning the community derived in the past from being productive. The following quote exemplifies the decrease in physical activity involved in everyday life:

Today you have everything in the house and everything is [readily] available. When you go into the supermarket, you find a million things, take a few things and sit in front of the TV for three or four hours before bedtime. In the past, people went to sleep early, got up early, while today you sleep and get up and eat many diabetes-[inducing] foods (Tawfiiq, a 52-year-old woman from a recognized settlement).

The following quote demonstrates the decrease in the sense of collective meaning:

Listen, there is a problem of house demolitions. We have children who won’t have houses to live in in the future. There are unrecognized villages. The state confines us and leaves our needs as a society unfulfilled. They won’t let us grow fruits and vegetables and derive self-satisfaction from production. In addition, we are dealing with an increase in the cost of living (Salim, a 46-year-old man from a recognized settlement).

Attending to the perspective of members of the Bedouin community makes it possible to identify barriers to coping that are currently not addressed or even perceived as such though the external prism of the biomedical approach. As seen above, some of the barriers are the experience of land dispossession, house demolitions, and a forced abrupt transition from a nomadic to semi-urban lifestyle.

#### 3.1.2. Social Inequality: Inaccessibility to Health Care

Various researchers have linked the unfavorable treatment outcomes of T2DM to a lack of knowledge among members of the Bedouin community of “silent” diseases that may not cause symptoms at first (such as different types of cancer or diabetes), e.g., [13,19,21], thus attributing the unfavorable outcomes to a characteristic of the community itself. However, even if there is knowledge of the “silent” diseases, pre-symptomatic diagnoses (e.g., during routine screening tests) necessitates the accessibility to healthcare services. The inaccessibility of healthcare services was a subject that arose in many of the interviews. We noticed three reasons for inaccessibility: economic issues, a lack of health services in the region, and an inaccessibility due to communication problems.

Many interviewees recalled how the need to constantly work without taking leave prevented them from having a medical examination even years after the onset of significant symptoms. They sought medical assistance only after symptoms hindered their ability to work. For example, Khaled (a 67-year-old man from an unrecognized settlement) recounts how, despite having significant symptoms for years, he did not seek a medical examination. When asked why this was so, he responded:

I don’t know, I worked for many hours, day and night, never taking a day off, it was like that for several years. I worked as a security guard, [did] many shifts. I didn’t sleep for more than two hours a day. I wanted to earn money.

In addition to the inaccessibility that stems from economic factors, various interviewees referred to a lack of availability of healthcare services in the areas where they resided. For example, they mentioned long waiting periods for appointments:

We have difficulties in relation to healthcare services. We have only one physician, one administrator, one nurse, and one pharmacist [in the town], which causes a lot of difficulties […] Getting an appointment requires a long waiting period (Abed, a 67-year-old man from a recognized settlement).

Interviewees from unrecognized settlements reported an even more severe lack of accessibility to healthcare services. They spoke of medication shortages, especially long waiting periods, and a lack of public transportation to medical facilities.

In order to cope optimally with T2DM, patients require knowledge about nutrition, physical activity, medical examinations and diagnoses, their eligibility for social rights, and more. Culturally relevant and accessible knowledge is key for both early detection and treatment as the participation in low-cost accessible lifestyle modification programs is one of the most effective indicators of successful treatment outcomes [1,2].

Many interviewees lacked such knowledge, as Sarab, a 53-year-old woman from a recognized settlement reported: “I have symptoms, but I didn’t understand these symptoms.” A lack of knowledge is related to a lack in resources that are suitable for the Bedouin community, such as Arabic language medical pamphlets, medical articles, and reports in the media, as well as culturally appropriate medical education workshops (e.g., on the topics of nutrition and physical activity), as the following excerpt demonstrates:

Why is the Bedouin community neglected regarding this issue despite the fact that we are number one in relation to the ratio of those with diabetes? We have the highest percentages of diabetes [in relation to other communities/ethnic groups] […]. We even lack publications in Arabic: all medical magazines are in Hebrew. Why don’t we have pamphlets in Arabic to hand out to diabetics to save time for the dieticians? Why not publish a pamphlet of lifestyle recommendations for diabetic patients so that every physician could hand it out to his or her patients? (Aziza, a 48-year-old woman from a recognized settlement).

The Arabic language is still an official language in Israel, but government officials are not obliged to translate documents to Arabic and do so only in specific cases (Arraf-Baker, 2019). Moreover, patient illiteracy is also associated with a lack of knowledge regarding diabetes and how to cope with it:

[How do you know about your rights?] I don’t, I can’t read, and I can’t write […] My physician told me that I was entitled to four hours a week assistance, but I asked for it twice and I was turned down. (Na’ama, a woman in her seventies [exact age unknown] from a recognized settlement).

I’m very afraid of taking shots especially because I can’t read or write I’m afraid that I will get confused and inject the wrong dosage. I’m afraid of the test I do with the glucometer as it is. … We need a solution for these patients, assistance in the form of someone who will inject them and support them. (Sarab, 53-year-old women from a recognized settlement).

The interview excerpts presented above demonstrate that the issues of poverty, social exclusion, and marginalization are critical to the understanding of late diagnoses and difficulties in coping. Even in the presence of stressful symptoms, some interviewees felt that they should not take leave from work to be tested, while others had no access to information in Arabic.

### 3.2. Unique Resources for Coping with T2DM

#### 3.2.1. The Traditional Way of Life and Traditional Foods

In coping with diabetes day-to-day, one of the most dominant resources drawn from local indigenous culture was the traditional way of life. Various interviewees talked about how they used traditional methods of preparing food and ate traditional foods as a resource for coping with diabetes:

There are many foods that I use to enhance my diabetes control. Cooked jerisha [wheat porridge], freekeh [roasted green wheat]. Foods that give energy, such as freekeh. Healthy foods. Food that strengthens my health, unlike fried food, salty pastries, and sweet foods that ruin my health … I had to rediscover the [traditional Bedouin] healthy foods that we have always eaten (Fatma, a 50-year-old woman from a recognized settlement).

The words “I had to rediscover the healthy foods” demonstrates how any culture at a given date and time has a wide repertoire of cultural practices that include food practices. The interaction of the community with the physical, social, and political environments determines which practices take place and which do not.

In addition to eating traditional Bedouin foods, various interviewees told us how they still practiced some form of traditional agriculture (even partially) and, in this way, managed to incorporate the physical activity involved in traditional agriculture and household chores into their everyday lives:

I see household chores as exercise for me. I raise chickens and goats. I get up for them sometimes. When a goat gives birth, every 24 h I go to see her and return. Even household chores. I see them as exercise, even if they [healthcare professionals] say they are not exercise. (Fatma, a 50-year-old woman from a recognized settlement).

#### 3.2.2. Traditional Practices of Healing

In addition to incorporating the physical activity involved in traditional agriculture and choosing healthy traditional foods, another resource related to indigenous culture is the cultivation of plants that serve as traditional herbal medicines. The following interviewee cultivates medicinal herbs and distributes them to friends free of charge. Cultivating traditional herbs allows her to combine a social activity with traditional healing practices:

I have herbal plants that I grow in my garden and boil, like hyssop. Everything is natural … I always liked having plants, but when I became ill I tried things and I learned what helps and what’s good [with diabetes]. [How did you learn this?] Using my own personal experience, I tried things and I saw how it affected me, I also give people herbal infusions. [Do you also give people medicinal plants?] Yes, I give them infusions to try. I have a cousin with diabetes and I give her herbs, not instead of pills, but if you maintain a healthy diet and combine pills with herbs it goes well together. (Hanin, a 67-year-old woman from a recognized settlement).

Hanin suggests that her friends use herbs in addition to, and not instead of, Western medications. Similarly, other interviewees who said they used traditional herbs said they saw them not as a replacement for Western medications, but rather as a supplement.

#### 3.2.3. Religion

Another dominant resource drawn from Bedouin culture is religion. The majority of the interviewees in our study explicitly stated that religious beliefs and practices significantly assisted them in coping with T2DM. The fact that participants employed religion in active coping, as we demonstrate below, is especially revealing in light of the fact that “minority in transition” studies tend to frame religion negatively as a factor that complicates coping and encourages passivity and fatalism, e.g., [13,19,21]. According to this view, religion encourages fatalism, which makes patients avoid taking steps to improve their health (ibid.). Our interviewees tell a different story about the role of religion in coping, adding to our understanding of how religion functions in this context. First, religious belief has a wider meaning than mere fatalism in relation to coping with T2DM. Many interviewees told us that religious beliefs improve their quality of life by instilling optimism in them and helping them to regularly incorporate physical activity in their everyday lives:

It’s not good to lose hope. Belief in God strengthens hope and therefore believing is a positive force, a force that may help you cope, and religious belief really improves your morale. (Aadel, 59-year-old man from a recognized settlement).

In the context of the biomedical model, healthy diet instructions appear to lack any cultural connection. In contrast, our interviewees’ stories demonstrate how individuals can employ various myths and cultural narratives to incorporate and practice useful techniques to cope with T2DM, such as “intuitive eating,” which is the ability to remain attuned to one’s body while eating, and to stop eating as soon as one feels satiated:

When the Prophet said “when eating one needs to make room for water, air and food,” [he] didn’t mean that one needs to eat until one is satiated. No, [one should] stop eating even if he is not satiated. In practice, I follow this advice because it makes me more comfortable; if I eat a lot, it keeps me from falling asleep at night. [Our] lifestyle has changed, but we have preserved some habits (Yasmine, a 52-year-old woman from a recognized settlement).

Yasmine draws on the local culture’s resource of the Quran to instill meaning into changing her eating habits, specifically in the direction of intuitive eating, which is developing the ability to be attuned to one’s bodily cues and sensations of hunger and satiation.

Local culture can be used as a resource to instill meaning into other health related practices, such as being physically active. According to the biomedical model, physical activity is practiced in order to keep the body healthy and not necessarily to achieve other goals (unlike spirituality in yoga, for example). In contrast, several interviewees described how they combined a routine of physical activity with their religious practice and, in doing so, managed to add personal and collective meaning to it. Ibrahim, a 62-year-old man from a recognized settlement reported: “Yes, every day I walk to morning prayer at the mosque one and a half kilometers and there and back. I try to walk regularly.” In this context, Yamana noted the following:

I pray, fast and observe religion. It helps me very much and calms me. I mean, if I didn’t fast, I’d feel that everyone was fasting [except me] and that I was weak. That’s why I fight my illness. The best proof [for that] is that now I fast, I fight my illness as vigorously as possible for my religion and for Allah. Because he who makes an effort for Allah, Allah will compensate him for everything. (Yamama, 50-year-old woman from a recognized settlement).

Beyond the possibility of taking part in different practices, such as fasting, religion provides different ways in which to contribute to the community. One participant learned to read as an adult and volunteers to heal people by reading verses from the Quran. Employing the Quran in a therapeutic manner enables her to creatively combine an emotional/supportive form of therapy with social activism in the community. Using religion as a healing resource, in this case by reading verses from the Quran, is a good example of a coping method that goes beyond the biomedical model:

Every night I read verses from the Quran and it really helps me. I even heal other people with the Quran. I heal many people who approach me and I read instead of asking someone else to read and I don’t charge for it. Thank God for everything (Ruan, a 48-year-old woman from a recognized settlement).

Beyond improving our interviewees’ sense of well-being, increasing their optimism, contributing to their healing, and helping them to incorporate physical activity into their daily lives, religion assisted them in coping with the social stigma that surrounds T2DM. The social stigma of diabetes is a major force that complicates coping with T2DM in general, and in indigenous communities in particular. For example, many participants described how shame and fear of the social consequences of being identified as T2DM patients make them postpone or even relinquish important practices, such as glucometer testing, so as not to be seen engaging in them. Nevertheless, some interviewees describe how religious belief help them cope with the stigma attached to T2DM. For example, Na’ama, mentioned above, explained, “[I reveal the fact I have T2DM] to all of my friends and relatives… Yeah, why should I be ashamed? It’s from Allah.”

To conclude, the interviewees’ perspectives and experiences illustrate that local culture has the potential to facilitate coping with T2DM in at least two ways. The first of these is by inspiring them to use coping strategies not currently acknowledged by those who espouse the biomedical approach, such as engaging in traditional healing practices, traditional agriculture, and embracing a traditional lifestyle. The second is by enabling them to give culturally relevant meaning to coping practices, such as changing their eating habits and being physically active.

## 4. Discussion

### 4.1. Starting from the Indigenous Perspective: The Importance of the Coping Experience

The vast majority of studies that have examined coping with T2DM among the Bedouin in Israel have looked at the community from the outside. From this perspective, elements of local culture are seen mainly as obstacles or factors that interfere with optimal coping. The transition from a nomadic to a semi-urban lifestyle is depicted as inevitable and tends to be generalized, as if such processes take place in different periods and geographical locations in an identical manner. The specific conditions under which the transition of the Bedouin community has taken place, and their own experience of it, are not addressed.

In contrast, the current study was conducted in close collaboration with the Bedouin community and a conscious decision was made to examine the issues at hand from the perspective of members of the Bedouin community (Etic). By attending to the insider perspective, we were able to hear different stories about the relationship between elements of local culture and coping with T2DM, as well as the experience of illness and coping with the illness itself.

### 4.2. Identifying “New” Barriers to Optimal Coping

Learning about the experience of members of the Bedouin community from their perspective allowed us to identify barriers to optimal coping that are currently not addressed or recognized though the external prism of the biomedical approach. Two such barriers to treatment are the experience of land dispossession and the rupture of caring relationships (Duwe, 2016).

Our interviewees described the way in which the forced transition to semi-planned towns undermined the community’s ability to cope with T2DM by making it impossible to sustain a lifestyle based on traditional agriculture. In the context of a lack of employment opportunities and extreme poverty, cheap and processed foods replaced local agriculture. Poverty and the inability to engage in agriculture means that young women are forced to work long hours in low-wage jobs (if they are able to find employment at all), while the elderly (or other dependents) are left without support or social ties. This situation has resulted in various forms of spatial separation, such as house demolitions, which also have also disrupted the community’s caring and supportive relationships.

The experience of transitioning from a nomadic to a sedentary lifestyle, as told by our interviewees, is not universal or inevitable. It occurred under specific conditions of external pressure, extreme poverty, a lack of suitable infrastructure, a lack of educational and employment opportunities, and experiences of discrimination and marginalization by other ethnic groups in Israel. It is therefore not surprising that, as other researchers have found, many interviewees experienced this transition as highly stressful, e.g., [39].

Examining issues from the perspective of the indigenous people themselves enhances our understanding of the barriers they face. For example, in interviewing Bedouin men and women about the perceived accessibility of healthcare, the researchers were able to break the concept of the inaccessibility of healthcare into three interconnected problems: the language barrier, cultural competency, and gender-based barriers [40]. Likewise, the stories of our interviewees suggest a more nuanced and complicated understanding of the “lack of awareness of the risk factors and complications of diabetes” often depicted as a significant barrier to treatment among indigenous groups, including the Bedouin community in Israel, e.g., [13,19]. By means of the interviewees’ stories, it is possible to break this general barrier down into three interconnected problems. The first of these is the inaccessibility to healthcare due to economic problems. In such cases, even if individuals are aware of their symptoms and complications, their reluctance to take leave from work prevents them from seeking care. The second is a lack of informational resources tailored to the needs of the Bedouin community, such as medical pamphlets, medical articles, and media reports in Arabic. The third is the existence of significant rates of illiteracy.

As our findings show, illiterate patients experience great difficulties and high levels of anxiety over the daily management of T2DM. For example, they cannot properly read glucometers or medication package inserts. Literacy is not usually perceived as an urgent problem to solve in relation to coping with T2DM among indigenous patients. Likewise, a basic understanding of Hebrew is often taken as a given, but among elderly Bedouin women a lack of knowledge of Hebrew may lead to missed appointments or many hours wasted as they cannot understand they are being called through an automatic waiting room call system [40].

### 4.3. Transitioning from Compliance with Treatment to Active Coping and Identifying “New” Coping Strategies and Methods

There has been a transition from viewing optimal ways of coping with T2DM as (relatively passive) adherence to medical advice, to the active and creative balance of medical advice and the individual’s valued activities and goals, often including what is called strategic non-compliance [24]. The concept of active coping has been central to this shift, as it has become widely understood that coping with a chronic disease cannot be reduced to adherence to medications or tests alone, but should be considered from the perspective of more holistic, alternative definitions of health and well-being [25,26]. The findings of the current study suggest that we should take the shift to active coping another step forward by changing the perception of what it means to (successfully) cope with chronic illness.

Making such a change invites us to recognize and learn from ‘indigenous knowledge systems’ [41]. By listening to how the interviewees use resources drawn from local culture, such as returning to certain elements of the traditional way of life of their community (e.g., by engaging in agriculture), or turning to traditional herbal medicine or religion, one can identify new ways in which to think about, and proactively cope with, T2DM.

Most studies that have examined the Bedouin community from the outside have related religion or religious faith to barriers to optimal ways of coping with T2DM, e.g., [13,19,20,21]. In stark contrast to this view, the majority of interviewees reported that religious faith was of great importance to them in coping with T2DM. Religious texts, practices, and beliefs gave new meaning to practices such as intuitive eating, changing eating habits, taking part in physical activity, and raising the collective awareness of the disease. Recognizing that disease management can have cultural and spiritual connotations is important in bridging the chasm between healthcare providers and indigenous people [41] and it is relevant to other ethnic communities. One can also think of organized walking to places of worship or organized physical activities inside places of worship as possibly culturally relevant resources for the promotion of physical activity in the ultra-Orthodox Jewish community, which also suffers extremely high rates of T2DM and poor outcomes [42].

In many cases, in their everyday coping with the disease, our interviewees combined life domains that are usually separated under the biomedical approach: religion, work, family, physical activity, and social activism. For example, the participant who learned to read as an adult and volunteers to heal people by reading verses from the Quran uses religion as a resource and a healing option while strengthening social ties in the community. Another example in combining religious practices with everyday coping, for example, controlling sugar levels in a way that allows religious fasting or the employment of religious beliefs to cope with social stigma. Creating small scale enclaves in which traditional agriculture can be practiced enables individuals to combine everyday physical activities with work.

### 4.4. A Multifaceted Approach to Promoting Health, Health Knowledge, and Health Equity: Bringing the Pieces Back Together

A multi-faceted approach to health requires bringing back together life domains that have been separated, at least in the context of health. For example, physical activity need not necessarily be a distinct activity, practiced in private spaces such as gyms (that are often expensive and inaccessible to different groups) in the evenings (often a time dedicated to social or familial actives) [40]. Initiatives that grant communities, families, and individual women small plots of land where they can engage in agriculture can increase opportunities for physical activity in a way that is attuned to the culture of the community. Similarly, encouraging religious leaders to speak about symptoms and coping with T2DM and other chronic diseases may also be useful in encouraging the integration of collective and individual religious practices with habits that support healthy lifestyles.

The movement towards cultural competence in healthcare is gaining attention and has been recognized both by researchers and policy makers [40,43,44]. A multi-faceted approach to health requires bringing back together life domains that have been separated, at least in the context of health. For example, this could include cookbooks that offer traditional dishes that are suitable for T2DM, or leaflets with recommendations for better food choices that take into account the availability of products and the local food culture.

A multi-faceted approach to health requires bringing back together life domains that have been separated, at least in the context of health and medicinal systems, rather than treating indigenous people only through the biomedical model [41].

## 5. Conclusions

Conducting the present study in close collaboration with the Bedouin community enabled us to hear different stories about the experience of T2DM and coping with it. The “insider’” perspective allowed us to identify barriers to coping that are currently not addressed by the literature, many of them directly related to the enforced transition to a semi-urban lifestyle, such as various forms of spatial separation as well as disruptions in the community’s supportive relationships. By listening to how the interviewees use resources drawn from local culture, we have identified new ways in which to think about, and proactively cope with, T2DM. In addition, in their everyday coping with the disease, many of our interviewees combined life domains that are usually separated under the biomedical approach: religion, work, family, physical activity, and social activism. Accordingly, a multi-faceted approach to health requires bringing back together life domains that have been separated, at least in the context of health.

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
