# Peer review of "“I Had to Rediscover Our Healthy Food”: An Indigenous Perspective on Coping with Type 2 Diabetes Mellitus"

_ijerph, 2021, doi:10.3390/ijerph19010159_

Round 1

Reviewer 1 Report

1. The introductory section should be summarized. There is a lot of information that it does not contribute to the scientific field.
2. The precise methodological delimitation process. It is necessary to write down the Ethics Committee that validated the study. It is also necessary to write down the research protocol in this section. As an inclusion criterion, a person diagnosed with DM2 in the last 6 months remains poor, it is necessary to point out some other criteria as risk factors, another laboratory criterion such as glycosylated hemoglobin. The age range can generate a bias in the sample.
3. The experiments, statistics and other analyzes that were carried out do not appear in the results section.
4. The conclusions are not presented adequately and are supported by the data, they must respond to the objectives set at the beginning of the article.
5. The article is presented in an intelligible manner and is written in standard English.
6. The research does not meet all applicable standards for the ethics of experimentation and the integrity of research, add it in the methodology section.
7. Article does not adhere to appropriate community standards and reporting guidelines for data availability.
8. Bibliography:
Quote number 8: delete Hebrew at the end.
Quote number 11: delete the ellipsis and add the rest of the authors of the paper.
Citation number 27: italic format used for the journal in the title of the article, modify.
Appointment number 37: if the access date appears in this website appointment that appears in the rest.

Reviewer 2 Report

This study uses 49 semi-structured interviews among Bedouin in Israel to examine the coping strategies for Type 2 Diabetes Mellitus.  This is exciting research on an important topic. I believe this narrative study contributes to a better understanding of diabetes and related health outcomes. As diabetes is a leading cause of morbidity around the globe, this study will improve our understanding of risk factors of diabetes which helps to take necessary measures to curb this issue. Despite support for this study, I do have some comments and concerns on this manuscript. 

The main issue with this paper is the lack of motivation about the research questions, which include why is understanding risk factors for diabetes is important. Could you please add a paragraph describing understanding diabetes risk factors are essential in this context? That will make your story more engaging. Please see the following manuscripts.

Tuso, P. Prediabetes and lifestyle modification: Time to prevent a preventable disease. Perm. J. 2014, 18, 88

Joshi, R.D.; Dhakal, C.K. Predicting Type 2 Diabetes Using Logistic Regression and Machine Learning Approaches. Int. J. Environ. Res. Public Health 202118, 7346. https://doi.org/10.3390/ijerph18147346

There are several typos throughout the texts. Please pay attention to these errors. 

Reviewer 3 Report

The manuscript (ijerph-1461312) entitled ““I had to rediscover our healthy food”: An indigenous perspective on coping with type 2 diabetes mellitus” is by Maya Maor, et al. This is a qualitative study, which should not be a paper in International Journal of Environmental Research and Public Health. The authors used the perspective of Bedouins themselves to explore the distinct challenges they face, as well as their coping strategies. The authors discussed physical and social inequalities, and unique resources to actively cope with T2DM in the minority Bedouin population of Israel.  They claimed a need to expand the concept of active coping to include indigenous culture-based ways of coping (successfully) with chronic illness.

Major Comments:

  1. The introduction section should be revised. The rate and number of type 2 diabetes in the minority Bedouin population of Israel should be included. In addition, these numbers should be compared with other ethnical groups in Israel to show the disproportionate as claimed by the authors. Last, references are needed.
  2. In the introduction section, there is no need to have sections such as 1.1 and 1.2. Both section 1.1 and 1.2 contain information not needed and relevant to the topic of the manuscript. For example, in the introduction section 1.1, there is no need use significant space to describe the difference between type 1 and 2 diabetes as this is not a textbook or news commentary. You can include the numbers and rates of type 1 and 2 diabetes in minority Bedouin population of Israel with references.
  3. Most of the section 1.2 should be in the discussion section. In the introduction, please introduce what have been done.
  4. Questionnaires and survey forms should be included.

Round 2

Reviewer 1 Report

Dear authors,

I appreciate the modifications made to the document.

Even so,
1) I consider the introductory section to be very long.
2) Methodology: in relation to the inclusion criterion, despite being a qualitative study, the criteria must adhere to an objective parameter and not to a subjective interview, according to international scientific application.
3) Results: qualitative studies, although no experiment is carried out, some require statistical analysis, and taking into account the measurement of data, we consider it important to develop some type of test such as means, deviations or variances.

Still, the article has improved considerably.

Author Response

Reviewer 1

  1. Comments:

Dear authors,

I appreciate the modifications made to the document.

Even so,
1) I consider the introductory section to be very long.
2) Methodology: in relation to the inclusion criterion, despite being a qualitative study, the criteria must adhere to an objective parameter and not to a subjective interview, according to international scientific application.
3) Results: qualitative studies, although no experiment is carried out, some require statistical analysis, and taking into account the measurement of data, we consider it important to develop some type of test such as means, deviations or variances.

Still, the article has improved considerably. The authors addressed all my concerns. I do not have any comments.

Response:

Thank you for your review. We are happy you think we have addressed all your concerns and do not have any more comments.

Reviewer 2 Report

The authors addressed all my concerns. I do not have any comments. 

Author Response

Reviewer 2

  1. Comment:

The authors addressed all my concerns. I do not have any comments. 

Response:

Thank you for your review. We are happy you think we have addressed all your concerns and do not have any more comments.

Reviewer 3 Report

The authors only addressed my comments regarding the introduction section.

I have asked the authors to include the questionnaires, the interview content. This is important for recording the scientific data. The questions have to be included.

Author Response

Reviewer 3

  1. Comment:

The authors only addressed my comments regarding the introduction section.

I have asked the authors to include the questionnaires, the interview content. This is important for recording the scientific data. The questions have to be included.

Response:

Following this comment, we have translated our interview guide.  We attach the interview guide as a separate file, an attachment to the main manuscript. As in semi-structured interviews, the guide was used flexibly so as to enable the interviewer to follow the flow of the conversation and the specifies themes of each interviewer. 
